# COVID-19 Pandemic and Equal Access to Vaccines

**DOI:** 10.3390/vaccines9060538

**Published:** 2021-05-21

**Authors:** Matteo Bolcato, Daniele Rodriguez, Alessandro Feola, Giulio Di Mizio, Alessandro Bonsignore, Rosagemma Ciliberti, Camilla Tettamanti, Marco Trabucco Aurilio, Anna Aprile

**Affiliations:** 1Legal Medicine, University of Padua, via G. Falloppio 50, 35121 Padua, Italy; danielec.rodriguez@gmail.com (D.R.); anna.aprile@unipd.it (A.A.); 2Department of Experimental Medicine, University of Campania “Luigi Vanvitelli”, via Luciano Armanni 5, 80138 Naples, Italy; alessandro.feola@unicampania.it; 3Forensic Medicine, Department of Law, “Magna Graecia” University of Catanzaro, 88100 Catanzaro, Italy; giulio.dimizio@unicz.it; 4Department of Health Sciences, Section of Legal and Forensic Medicine, University of Genova, 16132 Genova, Italy; alessandro.bonsignore@unige.it (A.B.); camilla.tettamanti85@gmail.com (C.T.); 5Department of Health Sciences, Section of History of Medicine and Bioethics, University of Genova, 16132 Genova, Italy; rosellaciliberti@yahoo.it; 6Department of Medicine and Health Sciences “V. Tiberio,” University of Molise, 86100 Campobasso, Italy; marco.trabuccoaurilio@unimol.it

**Keywords:** vaccination, equity, public health, COVID-19, COVAX

## Abstract

The COVID-19 pandemic has evidenced the chronic inequality that exists between populations and communities as regards global healthcare. Vaccination, an appropriate tool for the prevention of infection, should be guaranteed by means of proportionate interventions to defeat such inequality in populations and communities affected by a higher risk of infection. Equitable criteria of justice should be identified and applied with respect to access to vaccination and to the order in which it should be administered. This article analyzes, as regards the worldwide distribution of anti-COVID-19 vaccines, the various ways the principle of equity has been construed and applied or even overlooked. The main obstacle to equal access to vaccines is vaccine nationalism. The perception of equity varies with the differing reference values adopted. Adequate response to needs appears to be the principal rule for achieving the criterion of equity in line with distributive justice. Priorities must be set equitably based on rational parameters in accordance with current needs. The entire process must be governed by transparency, from parameter identification to implementation. The issue of equal access to vaccination affects the entire world population, necessitating specific protective interventions. In light of this, the World Health Organization (WHO) has devised the COVAX plan to ensure that even the poorest nations of the world receive the vaccine; certain initiatives are also supported by the European Union (EU). This pandemic has brought to the fore the need to build a culture of equitable relationships both in each country’s own domain and with the rest of the world.

## 1. Healthcare Inequality and the COVID-19 Pandemic

The COVID-19 pandemic has evidenced the chronic inequality that exists between populations and communities as regards global healthcare [1,2], attributable to various factors [3] including the varying availability of resources, due to socioeconomic reasons, to prevent and cope with diseases.

Inequality as regards the risk of contracting COVID-19 is due to the fact that certain individuals, communities, and populations are at a higher risk of contracting the infection or of contracting a particularly severe infection [4] than others. The first risk is linked to individual circumstances [5] (immunodepression), social circumstances [6] (community living, close living quarters, homelessness), and work-related circumstances [7,8] (frequent contact with the public, poor protection in the work environment) [9]. The second risk seems predominantly linked to biological conditions (pre-existing illnesses), but it is clear that prohibitive socioeconomic factors are also pivotal, such as limited access to diagnosis and appropriate early treatment [10,11].

As far as populations and communities are concerned, the two types of risk should be considered jointly; for individuals, however, specific factors must be taken into consideration [12]. In order to ensure protection, proportionate interventions are required with the quali-quantitative properties necessary for abating the inequality that exists between populations and communities at a higher risk of COVID-19 infection or of a particularly severe COVID-19 infection.

Vaccination, without doubt recognized as an appropriate tool for the prevention of infection, necessitates planning and implementation in line with the principles of equity.

The issue we intend to examine in this article concerns the equal distribution of anti-COVID-19 vaccine resources, firstly, on a global level as regards each nation, and, secondly, on a local level in terms of the specific properties of each community within individual national populations. Naturally, the immediate vaccination of the entire world population is unattainable; vaccination must by necessity be conducted in stages. Equitable vaccine distribution must take into account the inequalities present among the various populations. Therefore, our objective is to evaluate the criteria and rules that will enable the implementation of equity and whether policymakers have applied said rules, or whether some have chosen to adopt vaccine nationalism.

## 2. COVID-19 Pandemic and Equal Access to Treatment and Vaccination

With regard to the COVID-19 containment measures, access to treatment and access to vaccination are issues that demand careful assessment and decision making based on one of the recognized principles of bioethics, namely justice. In the initial phase of the pandemic, the insufficiency of available resources became apparent when compared with the treatment needs of patients, resulting in the need to determine more specific criteria in relation to access to certain treatments, especially that of intensive care [13,14,15,16,17,18]. The availability of the various vaccines posed certain problems concerning (1) proper administration to the world population and vaccine distribution methods to each nation on a global level, and (2) the order in which to vaccinate individuals and communities in individual nations. In both situations, the principle of equitable justice must be applied.

The general issue concerns which rules to set to identify those to whom priority access to treatment and vaccination should be given in order to obtain positive results for the largest number of people possible and to reduce the risk of spreading the pandemic. With regard to vaccination, giving priority access to those most exposed to risk will result in greater efficacy for the entire population [19]. 

As regards the principle of justice in connection with vaccinations, the first issue relates to equity in vaccine trial programs [20,21], and the second pertains to the agreements which make financing for research dependent on the supply of a certain number of vaccine doses [22,23]. The third issue concerns the distribution of the vaccines themselves. During the initial stages of vaccine allocation after approval had been granted by the regulatory body, availability was limited considering the plan to vaccinate the entire world population, and manufacturer lead times were uncertain. As a result, the ideal of immediate vaccination for everybody was unfeasible, necessitating a staggered vaccination approach in line with population brackets.

Similar to access to treatment when an unexpected insufficiency of resources arises, access to vaccination in the current situation of insufficient doses and vaccination personnel invokes the topic of distributive justice, identified by Aristotle as the socially just distribution of assets and resources (Et. Nic. V, 4, 1131 b 25). Aristotle propounded that, since justice is linked to equality, distributive justice requires equals be treated equally and unequals unequally, hence the *polis* should distribute benefits and burdens proportionately. Equality relates to the objective, not the process.

The key characteristic of the process (which must obviously be capable of reaching the objective) is proportionality. The most appropriate proportionality for reaching equality is equity. In this specific case, equitable vaccine distribution must be ensured in relation to the inequality of the various populations [24,25,26,27,28]. Since equity is time-sensitive, priorities must be set in proportion to the reduction of the risk of contracting COVID-19 for the greatest number of people in a given population at a given time. 

## 3. Equity and Vaccine Nationalism

In the context of international vaccine distribution, equity has to contend with vaccine nationalism, deemed justifiable by some in the circumstances based on the concept of limited national partiality. In the current conditions created by the COVID-19 pandemic, this presents as crisis nationalism. In that regard, vaccine nationalism in a limited form is considered by some to constitute one of the elements of justice [29]. A scenario is thus created in which certain governments will acquiesce to their constituents by hoarding the vaccine manufactured in their own country and others vaccines manufactured elsewhere [30]. Stockpiling is already taking place [31].

Some firmly oppose vaccine nationalism, which is neither ethically justified nor is it a valid method for protecting the long-term interests of countries who intend to employ it [32,33]. It is simply an instrument used in political power play [34], which must be counteracted [35]. 

In an attempt to reconcile the varying standpoints, some hold that no conflict exists between an equitable distribution of vaccines and vaccine nationalism provided that rigorous parameters are imposed upon said national partiality and provided that it is neither allowed to violate human rights nor exacerbate situations of poverty [36]. Nevertheless, governments do have both the right and duty to ensure their citizens priority access to a COVID-19 vaccine [37]. The prevalence of vaccine nationalism is likely to form the main obstacle to equitable access to vaccines [38]. 

## 4. Understanding the Bioethical Principle of Justice from the Perspective of Equity

The bioethical principle of justice can be considered from the perspectives both of interventions by individuals (by healthcare professionals and people in general) and healthcare policies implemented by governmental and administrative decision makers.

Firstly, let us consider issues arising from healthcare policies implemented by means of both international and national strategies. Our proposition is that healthcare policy interventions must reflect the principle of equitable justice. Therefore, those with the decisional authority must transcend the criteria of equality as a means and manifest their adherence to equitable processes, identifying the criteria and parameters to be used in ensuring such.

Using the outline proposed by Beauchamp and Childress [39], recorded in Box 1 with the addition of the last heading, the order of priority for vaccinations can be set by applying one of the theoretical rules—each following a different perception of equity—discussed below. These rules are valid both for individuals and populations or communities, and they should be adapted to the specific context. Though mostly mutually exclusive, some limited integration is possible.

Box 1Equitable Rules, from Different Perceptions of Equity.(a)To each person an equal share(b)To each person according to need(c)To each person according to merit(d)To each person according to specific role(e)To each person according to free-market exchanges(f)To each person according to the value of his image

The “to each person an equal share” rule is not applicable in this specific case in and of itself. In the specific context of vaccine distribution to the world population, it could be rendered “to each state according to its population”. It would, however, represent an expression of the principle of equality rather than of equity.

The “to each person according to need” rule is indefinite, since it has various further subdivisions: need for vaccine protection due to current diseases, immunodepression, biological age, environmental conditions both at home and at work. In short, the need may be associated with the degree of risk of contracting COVID-19. This raises questions as to whether to evaluate each need individually, whether a hierarchy can be established for each need and how that might be graded.

In terms of populations and communities, such a hierarchy is not essential considering the fact that, generally, when a population or community is affected by one of these needs, the others also become manifest cumulatively: low-income nations tend to have more marked global health needs and a higher risk of contracting the infection and of infection severity. The same can be said for communities of institutionalized elderly adults. 

The “to each person according to merit” rule means prioritizing those who play key roles in the management of a country. The criterion, designed to safeguard fundamental public functions, is indefinite, ambiguous unless defined in extreme detail, a harbinger of oligarchic notions, on the basis of which priority vaccine access would be guaranteed to institutional political representatives (for example, in parliament and government), scientists, technologists and academics, those who provide essential services, and to artists. If applied to the distribution priority by population, this rule would privilege countries that claim to have greater administrative, scientific and technological qualifications, which happen to coincide with the more socio-economically stable countries. An attempt (failed) was made to apply this rule when the Director of Health of the region of Lombardy officially requested that, in distributing the vaccine, consideration be shown for the fact that her region provides the largest contribution to the growth of the national economy [40]. 

The “to each person according to specific role” rule privileges those specifically involved in vaccination activities. It could be extended to include healthcare professionals involved in the diagnosis and treatment of COVID-19 and even further to all active healthcare professionals. This rule tends be confused with the need linked to the risk to which healthcare professionals are exposed and with the rule of worth, as they also qualify as scientists. On a general level, this rule could be used by countries who have made a greater contribution to engineering the vaccines in order to claim privilege. 

The “to each person according to free-market exchanges” rule sets forth, as regards international relations, the principle of the lawfulness of stockpiling and, on a personal level, privileges those willing to spend. Acknowledgement of this rule would confirm, implicitly, that the measuring unit of equity is money. The rule thus conflicts with the concept of equity in distributive justice and should be rejected. It has been cited here only because it was mentioned by Beauchamp and Childress.

The “to each person according to the value of his image” rule was introduced in some countries during the initial vaccination stages. Priority was given to certain well-known individuals to serve as testimonials to the validity of the vaccination process in a type of propaganda campaign, often choosing people who, based on their health needs, would have been vaccine priorities in any case. On occasion, people who were eligible due to their general worth or specific role were chosen. Resorting to social influencers or popular sports or theatre personalities is completely anomalous, as it would create an unfair preferential network. 

The following observations were derived from a comparison of these rules. The “to each according to his need” rule respects the concept of equity as a means of achieving distributive justice. The rule is valid for both populations and individuals. Certain aspects of the other rules also merit consideration. As regards individuals, safeguarding those with direct involvement in vaccination activities should be considered in a positive light, even though, in reality, these people may also be included with those in need due to the risk of infection given the extent and frequency of exposure. Moreover, prioritizing vaccine access to those with fundamental roles in the country has partial merit provided it is properly justified and accurately defined. 

In summary, as regards access to vaccination, equality (that is, vaccine administration for all) is a key objective for the population and may be defined as follows: the entire population must be guaranteed equal protection against infection by COVID-19 by means of a progressive vaccination campaign that takes into account the fact that vaccine administration is aimed at unequal communities and people, and that as a result, access must be prioritized equitably in line with the needs of such populations, communities and individuals.

Healthcare professionals are also required to apply the principle of equitable justice vaccination administration. The organization and planning of access to vaccination is not in competition with individual vaccinators. These professionals may thus find themselves in one of two situations: if the vaccine organization and planning follow the criteria of equity, the role of the professional is simply to adhere to such principles. Conversely, in the absence of such equity, two alternative but unsatisfactory options remain: proceed with the unfair vaccinations or refuse to vaccinate, deeming the vaccine incapable of timely administration and therefore letting it go to waste. Whatever choice is made, professionals should endeavor to adjust inequitable vaccine organization and planning with the tools at their disposal. 

## 5. Needs as an Expression of Risk

The Rawls [41] theory of justice has a bearing on the present topic which proposes the maximization of the prospects of the least well-off individual. It does not mean that achieving equality is always possible, but it does open the way to improvement for those most compromised. According to Harsanyi [42], even if all other individuals in society had opposing interests of the utmost importance, the interests of the most prejudiced would always be priority.

From a general viewpoint, needs are linked to the greatest prejudice: in this specific context, needs can be considered as an expression of the risk of contracting COVID-19. 

In general, risk (R) describes the possibility of an event, which can be measured as a product of various factors that indicate the probability of a specific event occurring (P) and the extent of the related damage (D). In calculating a risk, consideration must be given to the ability to prevent the event and contain any consequences (K) by means of training, information and organization. As illustrated in Box 2, by modifying Yates’ risk construction formula [43], the measurement of R is derived from the product of P and D in relation to K.

As concerns the risk in question, it is important to factor in that the extent of damage caused by the disease derived from COVID-19 may occur with extreme variations: it may cause a disease with greater or lesser severity and duration, potential deterioration of one or more pre-existing conditions, potential varying quali-quantitative after-effects, or death. In addition, the K factor is highly variable in correlation with the accuracy of the prevention and containment measures put in place and with the degree of information and training received by those exposed. 

In the context of a community, for example, deciding whether to consider people with a higher likelihood of contracting the infection as a priority or those who predictably will suffer greater harm is a complex process. Clearly, those who are affected by both elements of risk will be prioritized. 

In any case, the actual degree of risk must be taken into account, especially the extent to which it has been reduced as a result of specific initiatives or whether it remains unchanged due to that particular facility’s organizational lack in that area.

Similarly, the vaccine campaign, as refined as it may be, should not give rise to relaxation in prevention and containment measures, which become necessary at various times depending on the progression or decline of the pandemic. 

Ultimately, the factoring in of needs is an aspect of risk management and, in this specific case, places the emphasis on activities that result in an increase in the K factor.

The risk can be lowered by giving prioritizing strategies capable of implementation. In general, it is easier to influence the P factor rather than D. For example, the K factor may be increased by careful application of suitable infection containment methods, whereas it is more arduous to affect any of the factors that increase damage such as pre-existing conditions, immunodepression, and advanced biological age.

Box 2Calculating the Risk of Contracting COVID-19.                               R = P × D                                    KKey:P: probability of contracting the infectionD: damage (severity of the disease and after-effects;
deterioration of pre-existing conditions; death)K: ability to prevent infection and contain consequences

## 6. Equitable Vaccination: Resources, Planning and Transparency

The availability of several vaccines poses issues of unavoidable complexity, since they are unequal in terms of conception, evidence of scientific efficacy, administration methods and timeframes, known effects and their staggered arrival to market [44]. This complexity may create inequity which potentially weighs further on the inequalities present in the communities at risk. However, equal distribution of available resources may still be possible even without data published in accredited journals regarding certain vaccines used in South America, Asia, Africa and Australia [45,46,47]. The rule of need-based distribution still stands. Data on all vaccines should be taken into consideration when they become available, especially if they highlight the possibility for further efficacy-related inequality. 

In order to achieve equitable vaccination, international planning is required to ensure the equitable distribution of resources among the various nations, using criteria designed to enable control of the pandemic in the most effective manner and to provide indications so that equal access to resources is guaranteed in individual nations. In other words, if equity on an international level dictates that distribution be planned with regard for the various socioeconomic and general health situations existent in nations as a whole, similarly, on a local level, equity in each individual nation dictates that vaccine access be prioritized for communities with specific socioeconomic and health characteristics. There will be other specific characteristics within each country and community, not to mention individuals, to be taken into account when planning vaccine access details. What is required is a pyramid-type system with uniform rules that are progressively applied in line with the global needs of entire populations, the population of individual countries, communities and individual people.

The priority planning process must be governed by equity, using rational, need-based parameters. The entire process must be characterized by transparency [48], from the creation to the implementation of reference parameters, which, given their complexity, must be exhaustively explained, with no assumptions made. In addition, control systems can be created to ensure continual transparency.

## 7. Supranational Initiatives Based on Equity

Equitable access to vaccines is an issue that concerns the entire world population [49,50]. Although several governments have rolled out initiatives based on vaccine nationalism in order to protect their citizens, decisions have also been made regarding action plans to protect the global population. These initiatives, despite the imperfect implementation thereof, are based on equity. In September 2020, the WHO devised the COVAX [51,52] equitable vaccine access plan. The plan is co-led by the Global Alliance for Vaccines and Immunization (GAVI) [53,54], the WHO [55] and the Coalition for Epidemic Preparedness Innovations (CEPI) [56] and is one of the three pillars of the Access to COVID-19 Tools (ACT) Accelerator [57], launched in April 2020 by the World Health Organization, France and the European Commission. The ACT Accelerator promotes global collaboration with the objective of, inter alia, facilitating equitable distribution of tests, treatments and vaccines. COVAX is dedicated entirely to accelerating the development and manufacture of COVID-19 vaccines and ensuring said vaccines reach poor countries. The plan offers initial doses for 92 participating countries in proportion to the population of each individual country. Once distribution has reached 20% of the population, the next step will be to consider the risk profiles of each individual country.

From an EU [58] perspective, the European Commission seeks to “ensure that each country receives doses based on a pro-rata population distribution key” (p. 3), “putting in place a coordinated approach to the distribution of vaccines across EU Member States” (p. 4). This is to ensure “that all Member States will have equal access to the available doses based on their population size” (ibidem). 

However, the proportionality of the number of doses in relation to the population size is not sufficient to guarantee equitable distribution as it does not take into account the inequalities between the various populations and their respective needs such as the local severity of the epidemic, availability of prevention and treatment resources, and the level of information and education. It cannot be denied that this criterion contributes to the prevention of further inequality due to poor vaccine distribution and represents an attempt at promoting equity; however, equity is unachievable by the application of a single parameter (item), in this case, proportionality in line with a country’s population size. Further-reaching plans are required.

For example, consider the Fair Priority Model [30,35], which proposed placing the priorities on a scale for inclusion in the COVAX plan. Another equitable vaccine initiative was developed by the US National Academics of Science, Engineering and Medicine at the request of the National Institutes of Health (NIH), and by the Centers for Disease Control and Prevention (CDC) [59]. 

## 8. Factors That Affect Equitable Administration of Vaccines in Individual Countries

In managing vaccine access in individual countries, certain situations arise that interfere with the automatic application of the principle of equity. For example, in the context of residential communities for the elderly, a situation may arise wherein numerous members of that community refuse vaccination and/or suffer side effects. In such cases, the first step is to evaluate whether such interferences can be resolved by means of appropriate information campaigns to stem vaccination refusal and medical treatment of the side effects. In the event that these situations cannot be overcome, equity will be considered globally in relation to other viable interventions to protect these subjects such as increasing, where possible, protection measures on an individual and local level.

In addition, the need to provide the appropriate information to people with limited cognitive abilities, such as those in situations of need as a result of advancing age, may also prove difficult in relation to priorities. This may even occur outside residential communities. The potentially prolonged amount of time required to provide such information conflicts with the desire for expedited vaccination organization (obviously with respect for the safety and dignity of those involved). It is essential to pay close attention to these potential obstacles and find appropriate corrective measures by means of appropriate resources.

Similar attention must be given to the information provided to those who, for various ideological reasons, refuse vaccination. Equity necessitates that these people receive specific attention as they are disadvantaged from the outset due to their bias and unwillingness to listen.

A similar issue has been raised specifically concerning those whose judgment has been heavily influenced by events that have cast suspicion over the extent of the adverse effects caused by the AstraZeneca vaccine. Some countries, when said vaccine is used, have witnessed a reduction in the number of people attending appointments and the need, for those who have attended, to provide more in-depth information to allay their fears. 

It must be noted that these factors—predominantly concerning vaccine refusal, which, though often irrational, must be respected—which render equitable vaccine administration within individual countries more difficult manifested in the initial phase of the vaccine campaign, affecting the very people who should have been the first to be guaranteed protection. 

Furthermore, the use of IT solutions for issuing calls to vaccination, while useful, must be balanced with the awareness that they may prove ineffective as regards people who are not capable of using, or simply do not have, the appropriate technological means necessary at their disposal. It is essential that more easily accessible means of communication be used within the appropriate timeframe (not postponed). Exclusive use of advanced technology would potentially exclude the frail, precisely those who need protection.

The poor or advanced in years who live in less convenient areas may struggle to reach vaccination centers. In that case, it would be necessary to arrange appropriate transport services or to organize satellite vaccination centers, including in institutionally organized communities.

## 9. Conclusions

There are certain practical difficulties to the implementation of our proposal for equitable distribution of anti-COVID-19 vaccines, on the one hand concerning the potential for differing perceptions of equity and the various criteria and rules adopted in its application, and on the other hand, the extent to which vaccine nationalism is entrenched in the rich population and affects policymakers.

Aristotle’s concept of proportionality as a reference for the application of the principle of equity may be delineated in line with the health needs expressed by the populations of individual nations or by the communities within each individual nation. Anti-COVID-19 vaccination represents a tool by which to acknowledge (obviously not to solve) health needs and, as a result, control the risk of spreading of the pandemic. 

To that end, coordinated and transparent planning is needed, ensuring equitable vaccine distribution on an international level, taking into account the various socioeconomic aspects and general health of entire nations, and similarly enabling on a local level, within each nation, priority vaccine access for communities with specific health and socioeconomic needs.

There are several international projects which have applied a kind of mixed approach system, identifying nations in need and then distributing the vaccine in proportion to the population of individual nations. This method cancels out the specific needs of the various populations by putting them on the same level, limiting the proportionality factor of anti-COVID-19 vaccine distribution to the nation’s overall population.

Basing intervention models solely on needs is a difficult undertaking for affluent nations. In fact, some wealthy nations have not even been able to organize interventions to support and facilitate vaccine access for communities with specific health and socioeconomic needs within their own borders.

The principle of equity is uncommon in affluent populations and governments. In fact, international relations betray the belief that rights are proportionate to the population and that those with the financial means can legitimately hoard essential resources.

This calls for a change in the predominant culture in rich countries. The immense suffering that has resulted from the COVID-19 pandemic, which has affected populations who are accustomed to healthcare security, may raise awareness in such populations of the existence of common fundamental needs on a global level and of their responsibility to step up and adopt approaches that will ensure adequate solutions for populations that lack sufficient autonomous resources.

As regards individual countries, populations and governments seem to have a better understanding of and to have adopted the principle of equity in the context of individual people and communities. Nevertheless, it appears that insufficiently compensated inequality persists, at least as far as Italy is concerned, which is perhaps attributable to inexperience and administrative incompetence. Those with positions of responsibility are being pressured to remedy such inequality by public opinion—an opinion seemingly motivated by the principle of equity despite the varied viewpoints regarding which parameters to adopt in order to enable coherent application of said principle.

## Data Availability

Not applicable.

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
