# Peer review of "COVID-19 Pandemic and Equal Access to Vaccines"

_vaccines, 2021, doi:10.3390/vaccines9060538_

Round 1

Reviewer 1 Report

With this communication, the authors would like to underline the inequality existing worldwide for accessing to vaccination in the context of the COVID-19 pandemic despite some efforts done by the WHO and the European Union. They presented different factors/key parameters to take into account to reach equity to have access to treatment and vaccination whatever the location of the country/region in the world. It is a very complex subject mixing several parameters (economy, nationalism, different perceptions of equity, availability of products, awareness…), at individual and global levels. In this paper, the authors tried to explain why it seems so difficult to reach the equal access to vaccination between countries and highlighted the fact that is also difficult to have equality at a country or regional level. Nevertheless, this equal access to vaccination and treatments worldwide is the key to succeed to fight against the disease and to control it in a proper way.

General comment: the authors shall highlight the "best solutions" to reach an equal access to vaccination according to the different parameters. This information is dispersed in the different paragraphs of the communication and most of time difficult to identify.

Lines 24-26 : This sentence is unclear for the reader. It shall be improved, maybe by doing two sentences instead of one.

Lines 130-132 : The authors should described what are the modifications.

Three references are not included in the text : 50, 51 and 52. They shall be removed from the list of references

It would be greatly appreciated to finish this communication by a brief summary/conclusion.

Author Response

we want to thank the reviewer for his work and his comments which allowed us to improve the manuscript. We have tried to resume the considerations dispersed in a summary way at the beginning of paragraph 9 in order to better focus on the "best solutions" suggested by us.

Lines 24-26 : This sentence is unclear for the reader. It shall be improved, maybe by doing two sentences instead of one.

We have made the text more explicit by eliminating the previous sentence and inserting three new ones.

Lines 130-132 : The authors should described what are the modifications.

We have modified the text, bringing back the complete list of Beauchamp and Childress and specifying which item we have added.

Three references are not included in the text : 50, 51 and 52. They shall be removed from the list of references

We have included references 50, 51 and 52 in the text

It would be greatly appreciated to finish this communication by a brief summary/conclusion.

we have inserted the conclusions paragraph and extended paragraph 9

Reviewer 2 Report

Estimated Authors,

I've read with interest the paper from Bolcato et al, dealing with the significant topic of the equal access to COVID-19 vaccines. This theme is both consistent with the aims of Vaccines, and of certain interest to its readers.

In fact, this commentary is well written, and addresses the main points encompassed by this specific topic. However, in my opinion, a couple of minor improvements are required.

  1. Authors should report, in the main title, that the present article is a commentary rather than a research paper
  2. in the early stages of the report, the topic that will be addressed, and how such a themes will be addressed, should be clarified and reported. Similarly, section 9 (i.e. towards a culture of equity) should be reframed as a conclusion section, aiming to summarize available pieces of information.
  3. Some statements lacks of proper references. For example, Box 2 include an equation that (to me) seems a reformulation of Yates risk construct, but a proper reference must be included. 

Author Response

I've read with interest the paper from Bolcato et al, dealing with the significant topic of the equal access to COVID-19 vaccines. This theme is both consistent with the aims of Vaccines, and of certain interest to its readers.

In fact, this commentary is well written, and addresses the main points encompassed by this specific topic. However, in my opinion, a couple of minor improvements are required.

Authors should report, in the main title, that the present article is a commentary rather than a research paper

we want to thank the reviewer for his work and comments that allowed us to improve the manuscript. We have indicated that the article is a commentary.

in the early stages of the report, the topic that will be addressed, and how such a themes will be addressed, should be clarified and reported.

At the end of paragraph 1, we have reported our goals more clearly, specifying that it is a comment.

Similarly, section 9 (i.e. towards a culture of equity) should be reframed as a conclusion section, aiming to summarize available pieces of information.

Paragraph 9 has been reworked as Conclusions; a large first part of an annotated summary has been included.

Some statements lacks of proper references. For example, Box 2 include an equation that (to me) seems a reformulation of Yates risk construct, but a proper reference must be included. 

we have entered the correct quote and indicated the contents of Box 2

Reviewer 3 Report

The work entitled “COVID-19 Pandemic and Equal Access to Vaccines” by Bolcato et al. deal with a very urgent and significant issue, the access of vaccination to all. Considering the pandemic situation we are facing at the moment and the restrictions implement for vaccination (mostly based on age and occupation) it is important to notice the inequalities between countries. These authors have compiled a manuscript that is concise and to the point and that provides the necessary information to understand this subject. I believe this manuscript is extremely well done, clear and, most importantly, necessary. The authors did a very important job and it should be considered for publication as is.

Author Response

we want to thank the reviewer for appreciating our work

Round 2

Reviewer 1 Report

I would like to thank the authors to try to reply to all my comments/remarks and questions. The modified paper has been well improved comparing to the first version.

I have two last remarks:

Line 152 : The word “each” is not at the right place in the sentence “ to person each an equal share”. The authors shall correct it.

Line 383 : the authors shall add the word “hand” after “on the other”.

Author Response

thank's for your work.

Both changes were successful.